# Non-Driving-Related Task Engagement: The Role of Speed

Sean Seaman [1,*], Pnina Gershon [2], Linda Angell [1], Bruce Mehler [2] and Bryan Reimer [2]

1   Touchstone Evaluations, Inc., Detroit, MI 48202, USA; langell@touchstoneevaluations.com
2   Massachusetts Institute of Technology Center for Transportation & Logistics AgeLab,
    Cambridge, MA 02142, USA; pgershon@mit.edu (P.G.); bmehler@mit.edu (B.M.); reimer@mit.edu (B.R.)
*   Correspondence: sseaman@touchstoneevaluations.com

**Abstract:** Non-driving-related tasks (NDRTs) have the potential to affect safety in a number of ways, but the conditions under which drivers choose to engage in NDRTs has not been extensively studied. This analysis considers naturalistic driving data in which drivers were recorded driving and engaging in NDRTs at will for several weeks. Using human-annotated video captured from vehicle cabins, we examined the probabilities with which drivers engaged in NDRTs, and we examined the relationship between vehicle speed and NDRT probability, with the goal of modeling NDRT probability as a function of speed and type of NDRT observed. We found that tasks that contain significant visual and manual components, such as phone manipulation, show strong sensitivity to vehicle speed, while other tasks, such as phone conversation, show no effects of vehicle speed. These results suggest that there are systematic relationships between NDRT patterns and vehicle speed, and that the nature of these relationships is sensitive to the demands of the NDRT. The relationship between speed and NDRT probability has implications for understanding the effects of NDRTs on safety, but also for understanding how drivers may differ in terms of the strategies they employ to modulate their NDRT behaviors based upon driving demands.

**Keywords:** driver distraction; naturalistic driving study; attention; non-driving-related task (NDRT); road traffic; road safety

## 1. Introduction

The dynamics of driving a vehicle while performing a non-driving-related task (NDRT) is an area of critical importance for human-factors investigations of crash risk (e.g., [1–3]), user interface design (e.g., [4]), and behavior when automated support systems are engaged (e.g., [5,6]). Though several studies have shown an increase in driver distraction during various NDRTs (e.g., [7]), engagement in NDRTs is relatively common while crashes are still relatively rare [8]—this apparent paradox can potentially be better understood by looking at the interplay between NDRT behaviors and the conditions under which they occur. There are many factors that may influence drivers' decisions to engage in NDRTs in real world conditions, including traffic, weather, road type and vehicle speed, which has long been believed to be an important variable [9]. Speed can influence a driver's choice of whether to engage in a task at a particular moment, as well as what type of task may be appropriate for that moment. For example, [10], observing naturalistic engagements with infotainment systems, found that, while 50% of the interactions observed were shorter than 2.2 s, longer interactions tended to occur when vehicles were stationary. Additionally, the task itself may in turn have effects on speed that the driver continues to travel, as well as other aspects of driving performance (e.g., lane position, time-headway).

Morgenstern et al. [11] summarizes a host of literature showing that drivers generally increase following distance, reduce maneuvers (such as lane changes), and reduce speed when engaged in NDRTs. Tivesten and Dozza [12] found that drivers adapted their NDRT behavior to driving conditions, such as strategically initiating visual-manual tasks after turn maneuvers. Other studies (e.g., [13]) showed that drivers do not engage in demanding

visual-manual tasks in difficult driving contexts, such as bad weather, and do engage in such NDRTs in low demand driving contexts, such as when stopped at a signalled intersection [14].

In an analysis of naturalistic data from the second Strategic Highway Research Program (SHRP2), Risteska et al. [15] observed that driving environments do affect NDRT engagements as a whole, especially for older drivers. Examining baseline data (epochs from SHRP2 not containing crashes or near-crashes), the authors observed that increased speed (as well as increased environmental complexity, utilizing variables such as traffic level of service) was associated with fewer NDRTs, as well as shorter off-path glances. Utilizing the European Naturalistic Driving and Riding for Infrastructure and Vehicle Safety and Environment (UDRIVE) data, Ismaeel et al. [16] looked specifically at NDRT likelihood at intersections, with NDRTs being more likely to be observed when the vehicle was stationary than when moving, and when at an intersection controlled by a traffic light than a traffic sign. These studies both suggest that drivers engage in a level of self-regulation based on the driving context. Together, these behaviors operate in a dynamic fashion between driving context, driving behavior, and NDRT type.

The relationship between speed and NDRT engagement has been studied principally in terms of the distribution of on- and off-road visual attention. Senders [17] observed that diverting attention from the forward roadway disrupts driving more at faster speeds than at lower speeds. This is the case since, in manual driving, drivers are required to maintain moment-to-moment lane position and distance to lead vehicles, as well as engage in object and event detection, and visual attention cannot be diverted from the forward roadway for long, as the time-course of events requiring visual attention is reduced as speed increases. In line with this, interactions that involve greater investment of visual resources, such as traditional visual-manual human–machine interface (HMI) interactions, have been found to have a greater impact on driving speed than voice-initiated interactions [18] (e.g., there is a tendency for many drivers to slow their speed when engaging in a visual-manual task).

The relationship between visual attention and driving safety is inscribed in several standards documents of in-vehicle HMI design, including those of trade associations (e.g., Alliance of Automobile Manufacturers [19]; JAMA [20]), NHTSA's distracted driving guidelines for in-vehicle interfaces and aftermarket devices ([21,22]), and the European Statement of Principles on Human–Machine Interface ([23]). While the foundations of these guidelines are designed to mitigate the influence of task demands on driving safety, they are somewhat limited in their consideration of factors involved in a driver's decision to engage in NDRTs in real-world conditions. They are intended to be applied across all contexts, despite driving demand fluctuating wildly. This may lead to employing thresholds that are too permissive in high demand contexts, and too restrictive in low demand contexts. Understanding how drivers tend to safely approach engaging in different NDRTs across different driving contexts is a step toward better understanding what safer attentional patterns look like.

The current study evaluates the relationships between speed and engagement in NDRTs by leveraging real-world driving data. We examined the engagement in an array of typical NDRTs, categorized by their type and modality, across the spectrum of speeds from stationary to free-flow highway driving. We chose to do this for manual driving, despite the increased availability of partial-automation and the growth of research in automation, for two reasons. First, full automation, by some expert accounts, is "a transformation that is going to happen over 30 years and possibly longer [24]" and it is likely that, for the time-being, control of vehicles will likely be principally the responsibility of human drivers. Second, even with automation, NDRT engagement will likely affect the responsiveness of potential drivers (either in the vehicle or remotely) to take-over during transfer of control requests or silent automation failures. Thus, understanding the propensity of NDRT engagements across driving contexts in baseline manual driving can lay the groundwork for modeling distraction risk across different levels of automation. Different types of NDRTs have different components of demand from a driver, such as visual (e.g., looking at an

instrument cluster), visual-manual (e.g., mobile phone manipulation), or auditory-vocal (e.g., having a hands-free mobile phone conversation), and these types have been studied extensively in terms of distraction and driver crash risk (e.g., [1]). We hypothesized that the likelihood to engage in visual-manual NDRTs, such as smart phone manipulation and interaction with the center stack, would show greater sensitivity to vehicle speed compared to tasks that mainly rely on voice interactions, such as hands-free phone conversations and voice-based HMI interactions.

The goal of this study was to quantify the relationship between vehicle speed and the likelihood of engaging in NDRTs. Below, we describe the naturalistic study from which the data were drawn; the coding approach used; the analytic approach used; characteristics of the models relating NDRT likelihood to speed; and conclusions and implications for research and designers.

## 2. Materials and Methods

### 2.1. Participants and Data Collection

Drivers were recruited from the greater Boston Massachusetts area via flyers, social networks, forums, online referrals, and word of mouth. Drivers were screened using background and driving record checks, and were asked about driving habits to ensure that highway driving was a part of their regular commute. Drivers were compensated for their time involvement in the study with the use of a vehicle, one tank of gas, coverage of roadway tolls for the duration of their use of the vehicle, and a small monetary compensation. Twenty participants, evenly balanced by gender and with an average age of 54 years (range 22 to 66 years, sd 14.48 years), comprised the sample considered in this analysis.

This study was part of the ongoing MIT Advanced Vehicle Technology (AVT) naturalistic data collection effort (see [25] for additional details). Participants were provided one of two different MIT-owned and instrumented vehicle makes and models (2016 Range Rover Evoques and 2017 Volvo S90s) to use for one month and drive as they would their own vehicle. Periods of time during which partial automation (like ACC or lane centering) were active were excluded, as they were not generalizable to driving under full manual control or potentially even using other implementations of these same technologies. Initial analyses indicated that the vehicle type did not have significant association with the measures of interest, and accordingly, the following analysis did not include vehicle-type. Critical measures were obtained from cabin videos of drivers (coded as described below) and vehicle speed (collected from the controller area network bus (CAN-bus)). Drivers were aware their driving was being recorded, and were told they could request video to be deleted from the dataset.

### 2.2. Data Coding

Video recorded continuously during driving from two 720p (30 fps) cameras aimed at drivers (faces and seat/cabin area) was used for manual annotation of NDRT activities. Initial video coding was a collaborative, iterative effort involving senior staff and video analysts collectively developing a set of codes and definitions based on a subset of data. Subsequently, all analysts had hands-on training and received feedback from senior staff, and small portions of the data were dual-coded and assessed for inter-rated reliability (IRR), but no formal IRR was computed due to the large amount of data requiring coding. Video analysts identified and annotated NDRT engagement, including start, end, and type of NDRT for all periods of time when participants were observed manually driving (i.e., without semi-autonomous convenience features) on public roadways. The type of tasks coded were restricted to reflect tasks involving handheld devices (including mobile phone holding, manipulation, handheld and hands-free conversation) and specified vehicle HMI interactions (including center stack, steering wheel button, and voice-based interactions). This focus was based on theoretical relevance (HMI- and phone-related NDRTs remain of principle concern to distraction researchers and legislators alike), and implication relevance (potential implications of this work include HMI design decisions, which are

likely best informed by coding both HMI usage and smart phone usage, as both share similarities in technology, task demands, and driver motivations).

Tasks were coded as continuous unless there was a 5 s or greater pause mid-task, in which case tasks were ended at the last touch (for visual-manual tasks, such as phone manipulation, center stack HMI interaction, or steering wheel button HMI interaction) or indication of the end of a phone call (for handheld and hands-free phone conversation tasks). Phone holding included all times that a driver was holding a smart phone, but not using it for a handheld or hands-free conversation, or otherwise interacting with the device (drivers were not observed using phone holder devices in this sample, so all interactions with phones involved some degree of phone holding). Phone manipulation tasks included all types of smart phone interactions, such as browsing, dialing, or texting. HMI interactions included stereo, climate control, navigation, paired phone, or other types of tasks, and were coded by modality of input. Voice-based tasks were coded as subtending the period between the pressing of a steering wheel button associated with voice-task initialization, and the attainment or failure of the functionality associated with the intended voice command (e.g., display of route guidance instructions). All remaining NDRTs were labeled as "Other." However, because these tasks are of varied modality (e.g., talking to one's self is a primarily auditory-vocal task, while eating is primarily visual-manual), this category was not included in the analyses of task type, but overall percentage of time spent engaged in these "other" activities is included.

Manually-coded NDRT data were synchronized with vehicle speed data from the vehicle CAN bus network, which were recorded at 30 Hz. The entire dataset used in this analysis was subtended over 714 h of driving (about 35 h per participant).

## 2.3. Analysis Approach

The NDRTs were first examined by overall engagement propensity, operationalized as the probability of engaging in a given NDRT at any given time (Equation (1)), regardless of whether the task occurred alone or alongside other tasks, and regardless of vehicle speed. This is conceptually similar to other studies (e.g., [16]) that operationalize time spent on an NDRT (e.g., as a percentage of time), but is presented here as a probability for modelling purposes because, in subsequent analyses, the denominator will fluctuate based on speed.

$$P_j = \frac{\sum_{i=1}^{n} \frac{s_{i,j}}{s_i}}{n} \qquad (1)$$

where $P_j$ is the probability to engage in NDRT $j$ and $s_{i,j}$ is the number of seconds driver $i$ was observed engaging in NDRT $j$, and $s_i$ is the overall number of seconds driver $i$ drove. The total number of drivers in this study is $n$.

To evaluate whether NDRTs were performed differentially based on vehicle speed, a linear mixed-effects model with subject as a random factor was computed regressing NDRT likelihood against speed, NDRT type, and the interactions between these two factors. We fit a univariate model for smart phone NDRTs and a model for the embedded vehicle NDRTs.

Data were aggregated by participant and vehicle speed and grouped into five-mph bins. Speed bins ranged from 0 mph to 75 mph (preliminary analyses showed that driving above 75 mph was rare in this dataset and many participants had no observed driving above this speed). This yielded a table of 16 speed bins (zero, above zero to 5 mph, above 5 to 10 mph, up to above 70 to 75 mph, exclusive for each lower bound and inclusive for each upper bound) for each of the 20 participants in the study, with a propensity score for each of the seven primary NDRT categories (four smartphone-based NDRTs combined with three different modality vehicle HMI-based NDRTs). Propensities were computed similarly to Equation (1), with the number of seconds of observed NDRT engagement for each participant for each speed bin being divided by the total number of seconds of driving in that speed bin for that driver. Separate models assessing effects of gender and age did not yield significant effects for either, possibly due to small sample sizes within each category.

After visualizing and modeling linear effects, we also explored curvilinear relationships between NDRT propensity and speed, to develop stronger models of NDRT propensity and better reflect driver behavior. These were assessed on a task-by-task basis, comparing the goodness of fit of the linear model with the curvilinear alternative, and then re-modeling and plotting using the curvilinear transformation.

## 3. Results

The amount of data coded per driver varied between 10.9 h and 64 h ($\bar{x}$ = 35.8 h, sd = 17.2). The number of trips coded per driver varied between 29 and 117 ($\bar{x}$ = 89.8 trips per driver, sd = 37.8). The average trip duration was 23.9 min (sd = 25.4). Table 1 shows the breakdown of percentage of time spent performing each task. NDRTs were quite common, with 45.3% of driving time observed in this study being coded as containing a NDRT using the coding criteria described above, and 10.6% of driving time being coded as containing one of the NDRTs modelled in this study.

**Table 1.** Overall percentage of each non-driving-related task (NDRT) time being performed, expressed as a percentage of the total time of observable driving. Blank cells indicate combinations that were not coded, by definition (e.g., handheld phone conversations and holding phone) or for tabling purposes (e.g., "other" tasks were only assigned to the first task if a driver was not performing any other tasks during that time). "Manip" = phone manipulation; "Hold" = phone holding; "HH" = handheld phone; "HF" = handsfree phone; "CS" = center stack; "Wheel" = steering wheel controls; "Voice" = voice-based controls.

| First NDRT | Hold | HH Conv. | HF Conv. | CS | Second NDRT Wheel | Voice | Other | Multiple | None | Total |
|---|---|---|---|---|---|---|---|---|---|---|
| Manip. | | 0.03% | 0.03% | 0.006% | 0.005% | 0.01% | 0.47% | 0.01% | 1.51% | 2.07% |
| Holding | | | | 0% | 0.002% | 0.02% | 0.73% | 0.006% | 1.83% | 2.60% |
| HH Conv. | | 0% | | 0% | 0% | 0.001% | 0.19% | 0% | 0.51% | 0.70% |
| HF Conv. | | | 0% | 0% | 0.002% | 0.002% | 0.56% | 0.0007% | 1.90% | 2.46% |
| CS | 0.02% | 0.001% | 0.02% | 0% | 0.0005% | 0.005% | 0.49% | 0.02% | 1.36% | 1.92% |
| Wheel | 0% | 0% | 0% | 0% | 0% | 0.006% | 0.07% | 0.002% | 0.29% | 0.37% |
| Voice | 0% | 0% | 0% | 0% | 0% | 0% | 0.24% | 0% | 0.29% | 0.53% |
| Other | | | | | | | | | 34.64% | 34.64% |
| None | | | | | | | | | 54.71% | 54.71% |
| Total | 0.02% | 0.03% | 0.05% | 0.01% | 0.01% | 0.05% | 2.76% | 0.04% | 97.03% | 100% |

### 3.1. Engagement Propensity

Engagement propensity by driver is shown in Figure 1, where the black triangles correspond to the values computed using Equation (1). NDRTs are color-coded by modality, and three smart phone tasks—phone holding, phone hands-free conversation, and phone manipulating—were the most frequently observed of the NDRTs broken out for study. Additionally, Figure 1 suggests that the most commonly observed NDRTs—phone holding, hands-free conversation, and phone manipulation—also had substantial between-participant variability.

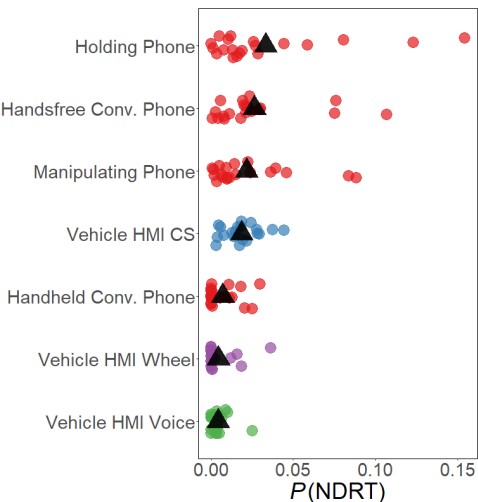

**Figure 1.** Task engagement probability as a function of modality. Circles indicate participant mean propensities, and triangles indicate mean group propensities. Circles are color-coded by modality, where "Phone" (red) indicates smart phone NDRTs, "CS" (blue) indicates center stack NDRTs, "Voice" (green) indicates voice-based NDRTs, and "Wheel" (purple) indicates steering wheel button-based NDRTs. NDRTs are sorted by overall mean propensity from highest to lowest.

*3.2. The Influence of Vehicle Speed on NDRT Propensity*

The first seven rows of Table 2 show the main effect and interaction terms for these models, in which likelihood of engaging in a NDRT at any given time was regressed against vehicle speed, NDRT type, and the interaction between these two terms in a mixed-effects linear regression using subjects as a random factor (overall conditional $R^2$ = 0.281). Ignoring which specific NDRT was being performed, there was no main effect of vehicle speed on NDRT engagement probability. In contrasting tasks with one another, we chose handsfree mobile phone conversation as the baseline comparison task, as it is the only pure auditory-vocal task in the group, and it is the most likely of all the tasks considered to not be associated with enhanced crash risk across a variety of crash risk computations (e.g., [2]).

**Table 2.** NDRT modeling by NDRT type, speed, and interaction. Model 1 ("All NDRTs") contains linear speed effects, NDRT type (contrasted with hands-free phone conversation) and the interaction between the two terms. Models 2 and 3 ("Man. Phone" and "HMI CS") contains reciprocal exponential speed effects. Models are all mixed-effects linear models with the driver as a random factor. Only statistically significant terms are shown. *B* indicates the unstandardized regression coefficient for that term; *t* is the ratio of the coefficient to its standard error; and the significance indicators (last column) indicate whether that ratio is significantly different than zero (all terms here were significant). "HF Conv." = hands-free phone conversation; "HH Conv." = handheld phone conversation; "HMI Voice" = voice-based interactions with vehicle human-machine interface (HMI); "HMI Wheel" = steering wheel button interactions with vehicle HMI; "Man. Phone" = phone manipulation; "HMI CS" = center stack HMI interactions.

| Model | Variable | | B | t | |
|---|---|---|---|---|---|
| | | HH Conv. | −0.0179 | −5.22 | *** |
| | HF Conv. vs. . . . | Holding Phone | 0.0178 | 5.22 | *** |
| | | HMI Voice | −0.022 | −6.53 | *** |
| All NDRTs | | HMI Wheel | −0.021 | −6.09 | *** |
| | | Holding Phone | −0.00041 | −5.26 | *** |
| | Speed X (HF Conv. vs. . . . ) | Man. Phone | −0.00050 | −6.40 | *** |
| | | HMI CS | −0.00017 | −2.16 | * |
| Man. Phone | | $1/e^{speed}$ | −0.00048 | −8.48 | *** |
| HMI CS | | $1/e^{speed}$ | −0.00015 | −5.22 | *** |

* indicates $p < 0.05$; *** $p < 0.001$.

Compared to hands-free phone calling, only phone holding was more likely to be observed, while handheld phone conversations, steering wheel button HMI interactions, and voice-based HMI interactions were significantly less likely to be observed. Phone manipulation and center stack HMI interactions were, statistically, equally likely to be observed among the participants as hands-free phone conversation. The interaction terms suggest that phone holding, phone manipulation, and center stack HMI interaction were statistically less likely to be observed as vehicle speed increased, while the other tasks showed no significant relationship to vehicle speed.

These relationships can be seen in Figure 2, which shows the average likelihood of NDRT engagement as a function of speed (solid lines) and the linear modelled likelihood of engagement (dashed lines) for each of the tasks under investigation. For hands-free phone conversation, handheld phone conversation, and vehicle HMI interactions via voice commands and wheel buttons, there is no relationship between NDRT probability and vehicle speed—the relationships are essentially flat. Hence, drivers were equally likely to engage in any of these tasks at a high vehicle speed as they were at low speed, with the caveat that each task has its own specific engagement probability. On the other hand, phone holding, phone manipulation, and center stack HMI interaction, become less likely as vehicle speed increased. Notably, these are tasks with physical and/or visual demands, as well as being the most likely to be associated with increased crash risk (e.g., [26]).

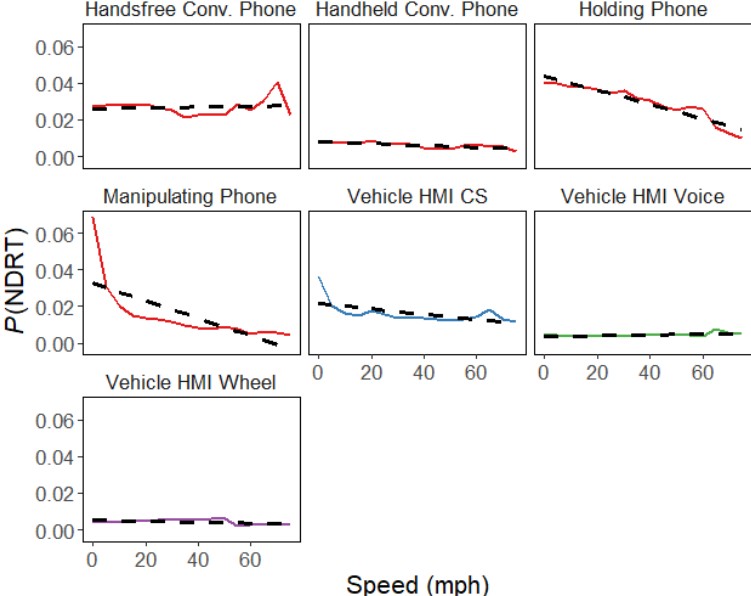

**Figure 2.** NDRT engagement probability as a function of modality, task type and vehicle speed. Solid lines indicate averages, and dashed lines indicate linear modelled values.

The curve shapes of two of the tasks that were associated with a significant relationship between NDRT probability and vehicle speed—manipulating phone and center stack HMI interaction—were suggestive that there exists a non-linear relationship between these two tasks and speed. We contrasted a model predicting each NDRT probability with a reciprocal exponential speed effect, defined in Equation (2):

$$P(NDRT \mid speed) \sim \beta_0 + \beta_1 \frac{1}{e^{speed}} \tag{2}$$

where *P*(*NDRT* | *speed*) is the probability to engage in an NDRT given being in a specific speed bin; $\beta_0$ is the intercept term; and $\beta_1$ is the coefficient by which the reciprocal of *e* (approximately 2.71828) raised to the power of vehicle speed is multiplied. This was done individually on a model containing only phone manipulation probabilities for each speed bin, and for a model containing only HMI center stack interaction probabilities for

each speed bin, and each was contrasted with the simple linear version of each model (for each model, mixed-effects regressions were computed). Reciprocal exponential speed effects are listed in Table 2 (bottom two models). For phone manipulation, the model incorporating a reciprocal exponential speed effect explained significantly more NDRT probability variance than the model incorporating a linear speed effect ($x^2 = 47.87$), with a conditional $R^2$ of 0.457. For center stack HMI interaction, the model incorporating a reciprocal exponential speed effect also explained significantly more NDRT probability variance than the model incorporating a linear speed effect ($x^2 = 39.43$), with a conditional $R^2$ of 0.524. Additionally, we applied Equation (2) to all the NDRTs together, and found that the non-linear speed term was a significant improvement over the linear speed term model ($x^2 = 3.79$), with a conditional $R^2$ of 0.121. Thus, not only does non-linear modeling better represent two visual-manual NDRT likelihoods as a function of speed, but it better represents the relationship between NDRTs and speed at large.

Figure 3 shows these two NDRTs, superimposed with a reciprocal exponential model of speed against NDRT probability. As is evident, this curve shape is notable for having a rapid change at low values of speed (near 0), and a near flat trajectory as speed increases. This is in contrast with phone holding (Figure 2), in which the probability of holding a phone changed linearly with speed, with all changes in speed throughout the range of speeds observed being associated with a consistent change in holding probability.

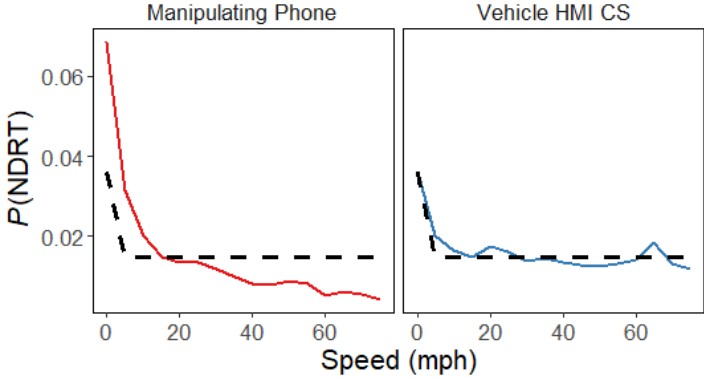

**Figure 3.** NDRT engagement probability as a function of task type and vehicle speed for phone manipulation and center stack HMI interaction. Solid lines indicate averages, and dashed lines indicate reciprocal exponential modelled values.

## 4. Discussion

The relationship between vehicle speed and NDRT engagement was examined and revealed that the probabilities to engage in NDRTs follow various speed-dependent patterns. Phone holding was the most common activity (of the set examined), although considerable between-driver variability was present. Holding, phone manipulation, and center stack HMI interaction NDRT types were strongly associated with vehicle speed. Phone holding showed a linear relationship, suggesting that drivers do not immediately stop phone holding as vehicle speed rises above a certain threshold, or vice versa. In contrast, a reciprocal exponential function provided a better fit when modeling the changes in the probability to engage in phone manipulation and center stack interactions across speed bins, suggesting that at a speed above ten miles per hour these activities become exceedingly rare.

This finding implies that the association between NDRT engagement and vehicle speed follows different patterns depending on the NDRT. Specifically, for NDRTs with strong voice-involved components (as opposed to significant manual-manipulation components), there were no speed dependent variations in the probability to engage. Thus, these NDRTs were as likely to be observed with the vehicle at standstill as they were in free-flowing highway traffic, and all speeds in-between. This was true for both handheld and hands-free phone conversations, as well as HMI interactions via steering wheel buttons and voice commands.

We hypothesize that the relationships between speed and NDRT engagement are traceable to one or more classes of driver behavior (as identified by [11]) including: 1. self-regulation, the practice of foregoing engagement in a desired behavior because conditions are understood to carry risk; 2. compensatory behavior, the adjustment of one behavior in order to engage in another (e.g., slowing down to text); 3. workload shedding, the removal of elements of driving or NDRT demand in order to accomplish very difficult tasks (e.g., reducing speed modulation in order to text); and 4. strategic engagement and management of tasks, the choice to engage in NDRTs when it is believed the risk of doing so is relatively low, given the driving context.

Accordingly, it is plausible that drivers engaging in NDRTs such as phone manipulation and center stack interactions are self-regulating or attempting to strategically manage task initiation by engaging in these tasks at lower speeds. These findings may reflect, in part, a tendency observed in experimental evaluation studies of similar task interactions (e.g., [18]) where compensatory behavior (shedding of speed) is observed when research participants are asked to engage with similar tasks. These potentially complementary findings suggest the importance of conducting observation under naturalistic conditions to fully place in context what is observed under experimental manipulation conditions.

The NDRT patterns observed could be seen as an indication of the amount of attentional capacity drivers had, or believed that they had, available based on the driving demands. As such, the patterns could support models of driver attention where attentional distribution should fit the demands of the situation. For example, Minimum Required Attention theory (MiRA [26]) posits that a driver is attentive if they are sampling sufficient information from the environment, to maintain a good representation of the environment, and should not be considered inattentive if this sampling is sufficient, even if the driver is engaged in an NDRT. The drivers in this study were frequently engaged in NDRTs, but this engagement appeared to depend, at least in part, on context, as operationalized by speed.

One potential application of these models is to adapt feature availability and modes of operation of HMIs based on driving demand. While the usage trends observed in this study hold in aggregate (for this participant sample), individual differences (both at the participant and trip-level) occur—instances where participants engage in NDRTs at ranges of speed where they are low probability events across the sample. By adapting HMIs to fit normative NDRT engagement patterns—for example, by adjusting limits on visual and manual demand in speed contexts where most drivers do not typically engage in such NDRTs—designers can encourage HMI usage that fits typical driver ability to manage task demands. This may be especially useful for novice drivers, who are less likely to be strategic in their engagement of NDRTs [27].

Though speed is an important factor in considering attention needs, it is of course just one of many factors that influence the immediacy, amount, and other characteristics of how attention is optimally distributed. While, all other things being equal, the relative risk of a sudden conflict tends to be higher at higher speeds, driving at high speed under semi-automated assisted driving on an uncrowded freeway in good weather is likely to have a very different risk profile than manually driving at relatively low speed on a crowded urban street in heavy rain, as would driving at similar speeds through road-types as varied as intersections, roundabouts [28], or alternative geometries (e.g., [29]).

Adaptive HMIs, based on driver real-time attentional needs, are an area of growing interest. Researchers and practitioners see real-time adaptive HMIs as an approach, in parallel with greater understanding of the driver's state, that can be used to mitigate risks associated with NDRTs and curb less strategic NDRT usage (e.g., smartphones vs. imbedded vehicles systems). Applications based upon this growing area of exploration, however, are limited by current voluntary guidelines (e.g., [21]) which include per se lockouts and certain other limits in a manner that is independent of driving context, and thus preclude application of the findings of this work and related research. Automation and other driver assistance systems that further augment the driver's role may amplify the need for adaptive limits based upon context.

There are several limitations to this analysis sample. While the drivers could engage in NDRTs where they chose to on roads they chose to drive on, they were driving assigned vehicles, not their own. Consequently, it is plausible that for most participants that the vehicle HMI was less familiar than that in their own vehicles, which may have impacted their likelihood of engaging in certain tasks or using certain modalities (such as voice-based tasks). While "Other" NDRTs were coded, they were too varied in demand characteristics to be meaningfully modeled in this analysis. Future research might benefit from a consideration of a wider range of NDRT categories through a more detailed annotation of the "Other" category to better capture the diversity of NDRT types and their prevalence of engagement. As computer-vision-based approaches to recognizing driver behaviors improve [30], coding the hundreds of hours of trip-level data from a project such as this at a higher degree of resolution will become possible with fewer resources. Additionally, because NDRTs were sometimes performed simultaneously (as shown in Table 1, nearly 3% of the driving time observed contained multiple simultaneous NDRTs), considering the additional demand of added NDRTs in the modeling effort could prove fruitful. Each moment of NDRT engagement was also considered equally, while its likely that as NDRTs subtend longer periods of time, they contribute more potential distraction; future approaches could weight NDRT engagement time points by how long a driver has persisted in the activity (perhaps using an algorithm like AttenD [31] to score engagement over time). Although some relationships between speed and NDRT likelihood were observed to be statistically significant despite the small sample, it is possible that other relationships could not be identified. This may especially be true for NDRTs observed less frequently. While manual coding of NDRT behavior at the trip level from cabin video is quite laborious, and thus increasing sample size is a non-trivial problem, it remains true that better identifying universal relationships between speed and NDRT likelihood would be improved by looking at a larger swathe of participants. In addition, drivers were aware they were in a study and were aware they were being recorded, which may have limited engagement in NDRTs, especially unsafe or illegal NDRTs. While this is also true for naturalistic studies that have evaluated the relationships between NDRTs and crash risk [8], it remains a limitation of instrumented-vehicle-based driving research. Finally, further consideration of the influence of environmental conditions and driver support features (e.g., ACC, SAE L2, etc.) are logical next steps.

## 5. Conclusions

NDRTs were not performed uniformly across the range of speeds observed, and the relationship between an NDRT likelihood and observed speed varied with the type of NDRT, with the demand characteristics of that NDRT (i.e., visual-manual) playing an apparent large role in that relationship. This suggests that drivers do show a class of highly predictable behaviors that likely mediate the relationship between certain NDRT probabilities and speed, although the nature of those behaviors—whether self-regulation, task shedding, compensation, or strategic NDRT engagement and management—is a subject for future study. It is our hope that these findings of patterns in NDRT engagement and vehicle speed will inform the development of future interventions to mitigate driver distraction.

**Author Contributions:** Conceptualization, S.S., P.G., L.A., B.M. and B.R.; methodology, S.S. and P.G.; formal analysis, S.S.; investigation, P.G., B.M. and B.R.; data curation, S.S., P.G. and B.R.; writing—original draft preparation, S.S.; writing—review and editing, S.S., P.G., L.A., B.M. and B.R; visualization, S.S.; supervision, P.G., L.A. and B.R.; project administration, L.A. and B.R. All authors have read and agreed to the published version of the manuscript.

**Funding:** This work was supported by the Massachusetts Institute of Technology Advanced Vehicle Technology (AVT) Consortium.

**Institutional Review Board Statement:** The study was conducted according to the guidelines of the of MIT's Committee on the Use of Humans as Experimental Subjects (COUHES) (protocol #1602469764R005).

**Informed Consent Statement:** Informed consent was obtained from all subjects involved in the study.

**Data Availability Statement:** Not applicable.

**Acknowledgments:** Data are from work supported by the Advanced Vehicle Technology (AVT) Consortium. The views and conclusions expressed are those of the authors and have not been sponsored, approved, or endorsed by the Consortium.

**Conflicts of Interest:** The authors declare no conflict of interest. The funders had no role in the design of the study; in the collection, analyses, or interpretation of data; in the writing of the manuscript, or in the decision to publish the results.

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
