# Peer review of "Non-Driving-Related Task Engagement: The Role of Speed"

_safety, 2022_

Round 1

Reviewer 1 Report

The study examined the relationship between vehicle speed and NDRT probability by analyzing naturalistic driving data from 20 participants, aiming to model NDRT probability as a function of speed and type of NDRT observed. 

Please address in the revision.

(1) Discuss what the appropriate number of the participants to model probability and why 20 is sufficient for the purpose.

(2) Articulate what this study "timely" contribute to the field. The study is about the NDRTs in non-automated vehicles (not automated ones!), which has been studied a lot for a long time.

Drivers are able to control the vehicle speed based what NDRTs they are doing; and they can also manage the workloads of NDRTs based on their vehicle speed to the best of their own abilities. In which applications would the presented function of speed and type of NDRT observed be "practically" incorporated in the HMIs? 

(3) Others

  • 43 - What does "half of interaction" mean?
  • 56 - What does SHRP2 project mean? What does UDRIVE mean?
  • 69 - The format (e.g., Sender [17]) is not consistently written whereas others are written as "Risteska et al. ...".
  • 84 - “While the foundations of ...” sounds the key motivation of this study. If so, it needs more in-depth discussion of the reasons why they are limited and why it is important to consider such factors.
  • 105 - Present the information of the standard deviation (APA styles?)
  • 105 - Wonder if you found any significant-different results by the factor of gender.
  • 107- It would be nice to explain how long the entire data collection period is (i.e., days, months) as a reference for those pursuing similar research methods.
  • 110 - The variance of two-vehicle settings might be on seat position considering two models are classified to whether its seat position is high (SUV) or low (Sedan). Right? Please describe more about why two vehicles were particularly used. For now, it's not sure why you didn't just intall the cam and OBD on the participants' own cars, and why the study used the two different MIT-owned and instrumented vehicles.
  • 120 - Describe how the video analyst group was trained and how many analysts were recruited for the study, with what criterion (if any).
  • 148 - The abbreviation was used for 'sd' whereas 'mean' is not. Be consistent and follow the format that the journal suggests.
  • 158 - There’s no description of abbreviation used in Table 1 (e.g., HH Conv., HF Conv., Manip., CS) while 'CS' is described in the latter part (line 180).
  • 168 - The variables (e.g., j, s, i, j) in this phrase may need to be italic.
  • 178 - Are circles a mean propensity? not a distribution of all participants' propensities?What do the colors mean? - read, blue, purple, green; better to describe it in the figure itself or its caption.
  • 187 - What the ‘uninvite model’ is? Everyone knows it?
  • 204 - Please check the table. It seems deformed and cropped, especially for the significance indicator.
  • 313 - Clearly describe and highlight the contributions of the current study. For now, it's hard to figure out and appreciate them.

Reviewer 2 Report

Dear Authors, thank you for sharing your analysis of data. I found your work very well presented. I have few comments that I would recommend to improve:

  • please elaborate more on the reasons why you seleceted the NDRT that you selected
  • please explain the labeling method a bit more. Who labeled the data and how did you approve quality?
  • the first table gave me some trouble. It needs improvement in formatting. At this point of reading I also could not allocate the abbreviations (e.g. HH Conv. or CS) with any of the NDRT. You may change the section of explaining the selected NDRT by adding a table of the selected tasks and their respectiv abbreviations. You may also use figure 1 as a legend (e.g. Holding Phone (Hold), Handsfree Conversation Phone (HH Conv.) etc. )
  • Figure 1 uses color-codes? I received it black and white and there is no difference in cycles. Also the resolution should be improved.
  • Table 2 is splitted over two pages. This is unrealable - also for a draft for review.
  • One of my biggest concerns is the relevance of knowing about any correlation between speed and NDRT. Please elaborate on this in the introduction and conclusion. Is there any application of this knowledge? Please discuss the benefit of your approach in comparison with driver monitoring and activity recognition. You may check and discuss in your introduction chapter those concepts:
    • Kircher, K., & Ahlstrom, C. (2017). Minimum required attention: a human-centered approach to driver inattention. Human factors, 59(3), 471-484.
    • Martin, M., Roitberg, A., Haurilet, M., Horne, M., Reiß, S., Voit, M., & Stiefelhagen, R. (2019). Drive&act: A multi-modal dataset for fine-grained driver behavior recognition in autonomous vehicles. In Proceedings of the IEEE/CVF International Conference on Computer Vision (pp. 2801-2810).
  • In spite of my doubt if speed is a relevant indicator to predict engagement in NDRT - I would apprechiate if you would discuss alternative indicators. Its not clear enought to me why you selected "speed". This is also a statistical issue. If we search for correlations among any possible data - we will find them. Even if its only by chance.
  • Please also discuss if your data is sufficient for your conclusion. You find a correlation between speed and visual-manual NDRT - but those are also the NDRT that had the highest occurance. May the other NDRT just have too little data?
  • Please explain why you decided to report the cumulated duration of NDRT and not the frequency of occurance. Short NDRT are underrepresented. There might e a reason for this (e.g. short is less risky) - please explain.
  • Please explain how you recruited the participants, how they were compensated and what form of informed consent they signed. For what kind of usage of their data did they agree and are they enabled to delete their data at any time?

I am looking forward to the revised verison andagain thank you for sharing your work.

Reviewer 3 Report

The manuscript describes a modeling approach to relate the probability of non-driving related task performance to the vehicle speed. The manuscript describes a useful and interesting analysis and it appears the modeling approach is sound. There are, however, a number of modifications that the authors should consider to make their manuscript more readable and increase the robustness of the analysis and its results. These are described in detail below.

  • The superscripts next to the author's names do not match the affiliations listed below
  • There is probably an overuse of "e.g." before references. I suspect there are many cases where the reference simply supports the statement being made, which does not require an "e.g."
  • The sentence between lines 31 and 34 is redundant with the points that are made in the next sentence, could be omitted.
  • In general, for the Introduction, there seems to be an important element of discussion missing. There are descriptions of several characteristics of the driving situation that may influence willingness to engage in an NDRT (e.g., intersections, weather, and, of course, speed), but there is little to no justification for why to focus on speed. Why did the authors pick speed from this set as the focus of this study?
  • Line 112: "significate" seems to be misused here. "significant"?
  • Lines 122-125. Why was this restriction put in place? What were the effects of the restriction (e.g., excluded a minimal number of NDRTs)? Actually, I see now that this represented ~35% of the time (Table 1)...which seems sizeable. More explanation is warranted.
  • Table 1 introduces the idea that tasks may have occurred concurrently...how was this implemented in the model? Was one task selected? If so, how was that selection made?
  • Lines 184 - 188. Equation 1 seems to be representative of the triangles in Figure 1. Correct? If so, any particular reason that the equation was not targeted for individual drivers? I'm mainly asking because in this paragraph the model is described with subjects as a random effect, which would suggest that the NDRT likelihoods that were input were those pertaining to each individual subject (and not the mean likelihood described in equation 1). Some clarification would be very useful. Also, the last sentence in this paragraph is confusing...what exactly is an "uninvite model"?
  • Should equation 1 include a consideration for speed bin? Or maybe another equation added that reflects the consideration of speed?
  • In general, certainly at the beginning of Sections 3.1 and 3.2 but also pertinent to other sections of the Results...there seems to be discussion of statistical approaches used to derive the results. These should be moved to the Methods section and the Results section should focus on the outcome of these approaches.
  • Lines 189-194 should definitely be in the Methods section and the equation referenced in the previous point may be appropriately referenced there. Furthermore, this is the first mention of a sampling strategy. Additional information about it is needed (prior to this statement, it appeared that all the video was reviewed and coded).
  • Line 190...reference to 16 speed bins, but there are different ways of defining those...please specify what the bins were a bit more clearly.
  • Table 2 is quite confusing. First, the table caption talks about differences of treatment of speed between models, linear in one case, non-linear in another. [Upon further reading, this seems to be explained in lines 239 through 242...this needs to be in the Methods section...see earlier comment to that effect]. This needs to be explained and justified in the main body of the manuscript. Furthermore, there are columns that are not clearly labeled, for example, what is "B"?
  • Line 198: "Considering all tasks together..." This suggest that there was a model created that was aggregated across tasks?  Was that the case? If so, it should be described? If not, please clarify.
  • Given the adjustments that had to be made to the treatment of speed for some tasks, why not go back and reconstruct the model with that information - adjusting the terms for these tasks to be non-linear?
  • Lines 280-272, starting with "suggesting..." Wouldn't the opposite be true (activities become exceedingly rare when speed exceeds 10 mph)
  • Line 329 mentions participant images in the paper, but none are shown.

Reviewer 4 Report

The content of the paper "Non-Driving-Related Task Engagement: The Role of Speed" could be interesting and within the scope of Safety. However, the analysis of the state of the art, the methodology adopted to investigate the problem, the proposed results are not compliant with the aim of a scientific paper. Figures do not have good quality. Table captions are very long and they are not proper captions. Table 2 is not complete. The results should be deeply analyzed. The reference sections does not comply with the template of the Journal.

The English language should be revised.

In my opinion the paper should be rejected.

Reviewer 5 Report

The topic presented in the paper is interesting and worth publication. In my opinion, the paper can be published, after taking into account the following remarks:

  • the text of the article should be transformed into an impersonal form (e.g.: done, presented, analyzed, etc.). At present, it is like follows: "presented, performed, added, received, etc.". It should be improved,
  • in the keywords section, the keywords "road traffic", "road safety" should be added,
  • at the end of Introduction section, the Authors should write what was the main aim of this paper, as well as what was contained in each paper section,
  • the Authors of the article discuss the impact of non-driving related task engagement, which affects speed. In the Introduction section, the Authors, present the circumstances in which such situations may occur, the infrastructure, and the conditions conducive to a non-driving-related task. Very good. However, the Authors should present their research problem in a broader context, referring to the issues of safe design of transport networks, designing safe solutions that increase the level of road safety as well as require increased attention from drivers. One of such solutions is designing roundabout intersections on the transport network, which have the above-mentioned features. Authors should refer to this issue and refer to the latest literature in this regard, e.g. "Roundabout entry capacity calculation-A case study based on roundabouts in Tokyo, Japan, and Tokyo surroundings, doi 10.3390 / su12041533", "Evaluation of Raised Safety Platforms (RSP) On-Road Safety Performance, doi.org/10.3390/su14010138" . One short paragraph in the Introduction section will be enough,
  • section "2. Materials and Methods" should not be divided into further sub-sections: 2.1, 2.2, and 2.3, because their content is too short (e.g. subsection 2.1 Participants consist of only two sentences). The Authors could either develop the content of each subsection (2.1, 2.2, and 2.3) or just not divide section 2 into further subsections,
  • Section "2. Materials and Methods" how many hours in total were gathered for further analysis? This information should be added in this section,
  • study participants were aware of the fact that they were taking part in the study. Did this fact have any influence on the results of the study?
  • the acronyms used in Table 1: the full names of these acronyms should be given (CS, HH, HF),
  • Table 2: The data inside table 2 are complete?
  • equation 2: the meaning of used variables should be added below the equation.

Round 2

Reviewer 1 Report

Thanks for the revised manuscript and author responses.

Reviewer 3 Report

The researchers have addressed my questions, I have no other comments. Nice work!

Author Response

Thanks!

Reviewer 4 Report

The paper can be accepted

Author Response

Thanks!

Reviewer 5 Report

The Authors have improved their paper according to the Reviewer's comments. Now, the paper can be published in a present form.